# Preparation and Characterization of Anti-Cancer Crystal Drugs Based on Erythrocyte Membrane Nanoplatform

**DOI:** 10.3390/nano11102513

**Published:** 2021-09-27

**Authors:** Lili Ren, Lirong Qiu, Binbin Huang, Jun Yin, Yaning Li, Xiaolong Yang, Guoguang Chen

**Affiliations:** 1School of Pharmacy, Nanjing Tech University, 5th Mofan Road, Nanjing 210094, China; yz1207cindy@163.com (L.Q.); huangbinbin97@163.com (B.H.); yinjun201962118004@163.com (J.Y.); captainya@163.com (Y.L.); cresc654123@163.com (X.Y.); 2Department of Microbiology and Immunology, Stanford University, Stanford, CA 94305, USA

**Keywords:** nanocrystals, erythrocyte membrane, drug delivery, antitumor effect

## Abstract

The simple and functional modification of the nanoparticle’s surface is used to efficiently deliver chemotherapeutic drugs for anti-cancer treatment. Here, we construct a nanocrystalline drug delivery system with doxorubicin wrapped in red blood cell membranes for the treatment of mouse breast cancer models. Compared with traditional free drug treatments, the biodegradable natural red blood cell membrane is combined with pure crystalline drugs. The nanoparticles obtained by the preparation method have superior properties, such as good stability, significantly delaying the release of drugs and enhancing the inhibitory effect on tumor cells. This study shows that the design of RBC as an outsourced drug delivery system provides a promising foundation for the continued development, clinical trials, and nanomedicine research of anti-cancer drug nanocarriers in the future.

## 1. Introduction

Nanoparticulate delivery systems are the preference for tumor medication in vivo attributed to their superior and unique properties, which can effectively improve drug bioavailability, serum stability, and drug metabolism [1,2,3]. In the pharmaceutical fields, the integration of complex functions to nanoparticles with simple methods becomes the optimal solution, and the functionality of surface modification of drug nanoparticles in vivo is expected to attain efficient delivery [4,5,6]. The use of natural biological materials (cell membranes or vesicles) as a carrier for nanoparticles is a viable functional modification strategy [7,8,9]. We do not need to perform these functions from the bottom up, since a series of characteristics of nanoparticles are endowed by the cell membrane, such as long circulation in the body, reduced uptake by the reticuloendothelial system (RES), and strong targeting [10,11]. In recent years, different types of cell membranes, including cancer cell membranes [12], macrophage membranes [13], platelet membranes [14], mesenchymal stem cell membranes [15], etc., have been exploited as carriers for coating various drugs for divergent therapeutic purposes, which enable anticancer drugs to be fully absorbed and utilized by the targeted tissues and obtain the ideal therapeutic effect [16].

As a kind of membrane carrier that was initially widely explored, erythrocytes can provide blood circulation for up to 120 days and have a relatively longer retention time than traditional synthetic encapsulation [17,18,19]. The immunomodulatory marker expressed on the surface of the red blood cell membrane is self-labeled protein CD47, which ensures that the macrophages in the body recognize it as themselves, thereby effectively avoiding the uptake of immune cells and achieving long-term circulation [20]. At the same time, 5 billion red blood cells per milliliter of blood make it the most abundant cell in the human body and have a high load capacity, which can be regarded as a well-stocked library of coating materials [21]. Furthermore, the double-concave disk-shaped mature RBCs can be deformed with the surrounding osmotic pressure. This critical swelling property makes RBCs an ideal carrier for coating a variety of biologically active substances, such as enzymes, proteins, drugs, active peptides, etc. [22,23].

The nanocrystalline drug is a kind of “pure drug particle” with high drug loading yields, usually in oral dosage form or injection dosage form [24]. However, many factors affect the stability of the nanocrystal system in drug delivery, such as particle surface area, formulation, external environment, and temperature [25,26,27]. Systemic medication may cause serious side effects, including capillary obstruction and embolism [28,29,30,31]. The treatment of nanocrystals against tumors is based on the prolonged circulation of drug accumulation, which requires protecting the surface of nanocrystals from the recognition of the reticuloendothelial system and the uptake of immune cells [32,33,34], thereby enhancing permeability and retention to reach solid tumors.

Thus, we endow the cell membrane coating technology to the drug nanocrystalline system, which consists of nano drug crystals as the core and the outer coating of the RBC membrane to form a membrane-coated nanoparticle with high drug-loading capacity and stable biocompatibility (RBC-NCs) [35]. Here, doxorubicin (DOX) was selected as an effective drug model, an insoluble chemotherapeutic drug with severe toxicity, to synthesize NC(DOX) and RBC-NC(DOX). Meanwhile, we demonstrate that the drug-loaded nanoparticles have a significant inhibitory effect on tumor growth. In vitro studies confirmed the long-term stability of phosphate-buffered saline (PBS), and the in vivo safety evaluation shows that the systemic toxicity of RBC-NC(DOX) is lower than that of DOX.

## 2. Materials and Methods

### 2.1. RBC Membrane Extraction

The red blood cell membrane is gathered by means of the previously reported method [36]. First, we collect fresh heparinized whole blood from male ICR mice (19–21 g) and centrifuge it at 7500 r/min for 5 min at 4 °C to remove plasma, white blood cells, and platelets. Then, the collected RBCs are washed with 1 × PBS until colorless and suspended in 0.25 × PBS at 4 °C for 30 min, and we remove the hemoglobin through centrifugation at 10,500 r/min for 10 min at 4 °C. Finally, the obtained erythrocyte membrane is purified with 1 × PBC and stored in normal saline at 4 °C for later use. All the animal procedures complied with the guidelines of the Institutional Animal Care and Use Committee at Nanjing Tech University, and the project authorization number is IACUC-20200507-01.

### 2.2. Preparation of RBC-NC(DOX)

Doxorubicin nanocrystals are prepared by the filming-rehydration method. Briefly, a certain amount of DOX (Shanghai Maclin Biochemical Technology Co. Ltd., Shanghai, China) is completely dissolved in dichloromethane by sonication. When the dichloromethane in the solution is evaporated to dryness through rotary evaporation at 40 °C, we vacuum dry the rest for 2 h. Next, we remove the dried DOX crystals from the glass wall and add them to distilled water to hydrate for 15 min and sonicate for 15 min. Finally, the solution containing the RBC membranes is mixed with NC(DOX) and ultrasonicated for 5 min; then, we make it pass through the 200 nm polycarbonate porous membrane.

### 2.3. Transmission Electron Microscopy Imaging

Transmission electron microscope (FEI, Hillsboro, OR, USA) is used to observe the morphology of NC(DOX) and RBC-NC(DOX). Briefly, a drop of particle solution (2 mg/mL) is dropped onto a glow-discharged carbon-coated TEM grid, which is followed by blotting after 5 min and staining with 10 μL of 0.75% phosphotungstic acid for 30 s. After the grid is dried, we observe the shape with a Talos F200X microscope (FEI).

### 2.4. Potential and Particle Size

The size (diameter, in nanometers) and zeta potential (ζ potential, in millivolts) of the RBC-NC(DOX) are measured by three dynamic light scatterings (DLSs) (Malvern Co. Ltd., Malvern Worcestershire, UK) with a Malvern ZEN 3600 Zetasizer.

### 2.5. Stability Study

The stability of the nanomedicine under storage conditions is verified by recording the change of the particle size of the nanomedicine at 4 °C. The nanomedicine is stored at 4 °C for 2 days, and the change in particle size is measured with a Malvern particle size analyzer (Malvern Co. Ltd., Malvern Worcestershire, UK) at 1, 2, 4, 8, 12, 24, and 48 h.

### 2.6. Aggregation

We take 0.5 mL of PBS and 10% FBS and add 0.5 mL of NC(DOX) and RBC-NC(DOX) solution. The OD value of the sample at the wavelength of 560 nm is measured on the microplate reader (Thermo Fisher Scientific Co. Ltd., Shanghai, China) at 0, 1, 2, 4, 8, 12, 24, and 48 h.

### 2.7. In Vitro DOX Release

We take 5 mL solutions of NC(DOX) and RBC-NC(DOX) respectively, and then dialyze them in 50 mL of 1 × PBS (pH = 6.5/7.4) dilution medium for 72 h at an ambient temperature of 37 °C and a rotation speed of 100 rpm. At a certain time point, 2 mL of release medium is taken to determine the content of DOX, and 2 mL of release medium is added at the same time.

### 2.8. Cell Culture

The 4T1 cells used in this study are from the Shanghai Cell Bank of the Chinese Academy of Sciences (Shanghai, China). The frozen 4T1 cells are resuscitated and cultured in a saturated vapor atmosphere of 37 °C and 5% CO_2_. The culture medium is RPMI 1640 medium containing 10% FBS and 1% penicillin. When the cells adhere to the wall and grow to 80–90% confluence, the cells are passaged and divided into plates until a sufficient amount of cells is obtained.

### 2.9. In Vitro Cell Uptake Experiment

Breast cancer 4T1 cells are selected as the model cells for the cell uptake experiment. The cells are prepared to contain about 1 × 10^5^ cell suspension per milliliter, added to a glass-bottom culture dish, and grow adherently in an incubator for 24 h. The original medium is discarded, and DOX and RBCNC(DOX) are added (the equivalent concentration of doxorubicin is 5 μg/mL) and incubated for 24 h. The supernatant is discarded, the cells are washed three times with 4 °C PBS to remove residual drugs, and then the cells are fixed with 4% paraformaldehyde, stained with Hoechst 33,258 dye for 15 min, and washed with PBS three times for 5 min each. We use CLSM (Leica Inc., Weztlar, Germany) to observe the uptake of cells and take pictures.

### 2.10. Cytotoxicity Assay

The in vitro cytotoxicity of free DOX, NC(DOX), and RBC-NC (DOX) on 4T1 cells is measured by the MTT method. In brief, the breast cancer 4T1 cells are seeded in 96-well plates. After 12 h of culture, the cells are treated with different concentration gradient drugs (0.1, 1, 5, 10, and 50 μg/mL) for 48 h, and the cytotoxicity is detected by the MTT method. The cell group without inhibitors is used as a control. At the end of the incubation, we add PBS solution and incubate them for another 4 h. Finally, we measure the absorbance of the cells in each well and analyze the cell viability.

### 2.11. In Vivo Safety Evaluation

The cultured 4T1 cells are digested with 0.25% trypsin. After the cells became round, we add culture liquid to stop the digestion, then centrifuge to discard the supernatant, add PBS to dilute the cells (1 × 10^8^/mL), and inject 4T1 cells into the right axilla of 6-week-old BALB/c mice; each mouse is injected with 100 μL. The inoculated mice will continue to breed until the tumor volume reaches 120–150 mm^3^ for in vivo experiments. The model mice are randomly divided into 4 groups (n = 6), namely Goup A: normal saline; Group B: free DOX; Group C: NC(DOX); Goup D: RBC-NC(DOX). They are administered via the tail vein on the 0th, 3rd, 6th, and 9th days without anesthesia, and the drug dosage is 5 mg of DOX per kg of body weight (5 mg/kg).

### 2.12. In Vivo Antitumor Efficacy

We set the first day of administration as day 0, measure the mice’s body weight and tumor volume every 3 days, calculate the tumor volume with the equation $V=\frac{{\mathrm{LW}}^{2}}{2}$, where V is the volume, L is the length, W is the width, and calculate the relative tumor volume. After a certain period of time, the mice are sacrificed, and the tumor inhibition rate of tumor-bearing mice in each administration group is calculated.

### 2.13. Histological Analysis

The heart, liver, spleen, lung, kidney, and tumor of tumor-bearing mice are taken out, soaked in 10% formalin solution, and normal tissues and tumors were gradually dehydrated with gradient ethanol. Then, after treatment with xylene, we embed the tissues in paraffin to make 5 μm sections and use xylene to remove paraffin from the tissue sections. Then, through gradient ethanol, the section is finally immersed in purified water for H&E staining. The stained sections are dehydrated by ethanol and then treated with xylene. After we drop the gum, the morphology of normal tissues and tumors is observed under a microscope.

### 2.14. Hemolysis Test

We prepare free DOX, NC(DOX), and RBC-NC(DOX) solutions with a concentration of 50 μg/mL. Then, 1.5 mL of the drug solution are taken into 1.5 mL of 2% red blood cell suspension and incubated in the water bath at 37 °C. After 3 h, the samples are centrifuged for 15 min. Then, 100 μL of supernatant are aspirated and added to a 96-well plate. The OD value at 540 nm is measured by a microplate reader (Thermo Fisher Scientific Co. Ltd., Shanghai, China).

### 2.15. Statistical Analysist

All of the results are presented as mean with SD. Data were analyzed using the Student’s *t*-test or one-way analysis of variance unless otherwise indicated. The two-sided *p* < 0.05 was considered statistically significant. Median survival times were compared using the log-rank test. Statistics were calculated using IBM SPSS STATISTICS 23.0 (Armonk, NY, USA).

## 3. Results and Discussion

### 3.1. Preparation of RBC-NC(DOX)

DOX nanocrystalline particles are prepared by the filming-rehydration method. Red blood cell membranes extracted from the blood of mice are coated on the nano-medicine crystal cores using the ultrasonic method described earlier. The general structure of the generated nanoparticles is shown in Figure 1, the DOX crystals are loaded in the core, and the RBC membrane coating and all related proteins form the outer layer.

### 3.2. Characterization

Preliminary observation through the microscope (Figure 2a) shows that the prepared RBC-NC(DOX) has a round shape and a relatively uniform distribution. We use transmission electron microscopy (TEM) to characterize the topography of NC(DOX) and RBC-NC(DOX). Figure 2b,c are the TEM pictures obtained by phosphotungstic acid staining. The heavy atoms in phosphotungstic acid are more capable of blocking scattered electrons than light atoms such as carbon, hydrogen, oxygen, and nitrogen in organic molecules. Therefore, the place where phosphotungstic acid is adsorbed (or deposited) looks darker in the TEM photo, while the place occupied by organic matter looks brighter. Generally, organic matter does not react with phosphotungstic acid, and phosphotungstic acid deposits around the organic matter to form a “background” or “contour”. As shown in Figure 2b, we can observe that the NC(DOX) is a regular spherical shape. A black outline will be formed, since the red blood cell membrane does not react with phosphotungstic acid. The image of RBC-NC(DOX) (Figure 2c) shows that the inner layer is the core of the nanocrystalline drug, and the outer layer is the red blood cell membrane. The nanoparticle size is about 200 nm. The TEM image can visually verify that we wrap the red blood cell membrane on NC(DOX), indicating that the method of synthesizing RBC-NC(DOX) is viable and effective.

In this experiment, a Malvern laser particle size analyzer was used to detect the average particle size and polymer dispersity index (PDI) of RBC-NC(DOX). It can be seen from Figure 2d that the particle size of RBC-NC(DOX) is normally distributed with an average particle size of 152.1 nm and a PDI of 0.180, showing the good control over the size of the nanoparticles.

In order to continue to characterize the RBC-NC(DOX), we synthesized that whether doxorubicin crystals have been covered with the red blood cell membrane. We measured the nanoparticle size of NC(DOX) and RBC with a nanoparticle size analyzer. As shown in Figure 2e, the average particle size of NC is about 141.8 nm, the average particle size of RBC is about 189.2 nm, and the average particle size of RBC-NC(DOX) is about 157 nm. The particle size of RBC is larger than that of NC, which helps NC to enter RBC more to form RBC-NC(DOX). RBC-NC(DOX) is slightly larger than NC(DOX) by a few nanometers, which shows that the red blood cell membrane has successfully wrapped on the nanocrystal through electrostatic adsorption.

Through the previous literature review, we learned that the surface of the red blood cell membrane is negatively charged. As shown in Figure 2f, the zeta potential of NC is about 1.58 mV, the average particle size of RBC is about −15.3 mV, and the average particle size of RBC-NC (DOX) is about −13.4 mV. The zeta potential of RBC-NC (DOX) is relatively close to that of RBC, and there is a significant difference from the zeta potential of NC, which proves the red blood cell membrane has a better package for the NC(DOX) synthesized in the early stage.

### 3.3. Physical Properties In Vitro

In order to evaluate the stability of our synthesized RBC-NC (DOX), the particle size is measured by dynamic light scattering (DLS). As shown in Figure 3a, RBC-NC(DOX) has a small particle size change under storage conditions, and the particle size is stable in the range of 150–160 nm. At the same time, NC(DOX) agglomerates fast, indicating that the RBC film coating can significantly improve the stability of NC(DOX). This also shows that our synthetic RBC-NC(DOX) has good stability in the blood and provides the possibility for intravenous injection of RBC-NC(DOX). 

In the previous research, we used the nanoparticle sizer to study the stability of the drug. While the protein in the serum is combined with the nanoparticles, it will cause the nanoparticles to cross-link and aggregate and result in a higher light scattering. Therefore, we evaluated the aggregation of the particles by measuring the absorbance at 560 nm in PBS (Figure 3b) and 10% FBS (Figure 3c). It can be seen that the RBC-NC(DOX) group has lower absorbance than the NC(DOX) group and remains stable in two days. In 10% FBS, NC(DOX) has a sharp increase in absorbance change. It is presumed that the nanostructure coated with red blood cell membrane effectively blocks the binding of serum proteins and nanoparticles.

In order to evaluate the stability of our synthesized RBC-NC(DOX) in different environments, we simulated two different in vivo environments. The pH 7.4 simulates the human blood environment, and the pH 6.5 simulates the lysosomal environment. The results are shown in Figure 3d. When the pH is 6.5, the average particle size of nanoparticles in 8 days gradually increases with time, and the stability is poor. It is speculated that the erythrocyte membrane coated on the surface of the nanoparticles is destroyed under this pH environment, which causes the aggregation of the nanoparticles. At pH 7.4, RBC-NC(DOX) maintained a relatively stable state for eight days. This also shows that our synthetic RBC-NC(DOX) has good stability in the blood, which provides the possibility for intravenous injection of RBC-NC (DOX).

We also studied the in vitro release kinetics of all preparations. Figure 3e,f are the drug release curves of NC(DOX) and RBC-NC(DOX) in different pH environments. We use the dynamic dialysis method to calculate the cumulative release rate of doxorubicin according to the standard curve of doxorubicin under different pH environments. The release behavior of NC(DOX) is similar at pH 6.5 and pH 7.4, while the release of RBC-NC (DOX) at pH 7.4 is slower than at pH 6.5. These results indicate that the stability and controlled release behavior of NC (DOX) are enhanced after the red blood cell membrane coated, and they show that the red blood cell membrane has a good protective effect on the nanoparticles in the simulated human blood environment.

### 3.4. Cellular Uptake and In Vitro Cytotoxicity of RBC-NC(DOX)

In order to evaluate the cytotoxicity of RBC-NC(DOX) drugs on the corresponding tumors, we conducted an in vitro cytotoxicity test. When free DOX, NC(DOX), and RBC-NC(DOX) are incubated with 4T1 mouse breast cancer cells in vitro for 24 h, the cell survival rates are shown in Figure 4a. In each administration group, the cytotoxicity was concentration-dependent, and the cell survival rate decreased as the concentration of doxorubicin increased. In order to compare the cytotoxicity of different DOX preparations, the IC_50_ value of each prescription preparation on 4T1 cells was calculated by SPSS (version 20) software (Table 1). For 4T1 cells, RBC-NC(DOX) has an IC_50_ value of 3.805 μg/mL, which is significantly better than free DOX with an IC_50_ of 5.587 μg/mL and NC(DOX) with an IC_50_ of 4.812 μg/mL. It shows that the drug covered by the red blood cell membrane can effectively inhibit the growth of tumor cells.

Since DOX has its own red fluorescence, we can visually judge the uptake of different DOX preparations by observing the DOX fluorescence intensity in 4T1 cells. It can be seen from Figure 4b–d that the red fluorescence in the cells of the free DOX group and NC(DOX) group was not strong, and a large number of uningested drugs are scattered around the cells. In contrast, the RBC-NC(DOX) group cells had strong red fluorescence, and there was no obvious drug scattered around, indicating that the drug-loaded nanoparticles were taken up by 4T1 cells. The observed differences in intracellular localization may be due to the absorption of nanoparticles through active mechanisms such as endocytosis [37], enabling them to have a higher transport capacity across the cell membrane than pure diffusion. Therefore, RBC-mediated drug-loaded nanoparticles facilitate the uptake of drugs by tumor cells, thereby improving the antitumor effect.

### 3.5. In Vivo Antitumor Effect on Subcutaneous Tumor Model

In order to evaluate the therapeutic effects of drugs in RBC-NC(DOX) on tumors in vivo, we conducted in vivo antitumor experiments on subcutaneous 4T1 mouse breast cancer tumor models. The growth and metastasis characteristics of 4T1 cells in BALB/c mice are very similar to breast cancer in humans. This tumor is an animal model of human stage VI breast cancer [38]. 4T1 cells were implanted subcutaneously into the right side of 6-week-old BALB/c mice, and the inoculated mice will be allowed to develop until the tumor volume reaches 120–150 mm^3^ for in vivo experiments. Model mice were randomly divided into 4 groups (n = 6), Group A: normal saline; Group B: free DOX; Group C: NC(DOX); Group D: RBC-NC(DOX). They were administered via the tail vein on the 0th, 3rd, 6th, and 9th days without anesthesia, and the drug dosage was 5 mg of DOX per kg of body weight (5 mg/kg). The volume changes of solid tumors after administration are shown in Figure 5a,b. Compared with the normal saline group, other preparations effectively inhibited the growth of tumors, and the RBC-NC(DOX) treatment group showed the significant effect in inhibiting tumor growth.

As shown in Figure 5c, the tumor inhibition rate of RBC-NC(DOX) was higher than that of free DOX and NC(DOX). Although not a powerful targeted formulation, it is believed that RBC-NC(DOX) can accumulate at the tumor site through enhanced penetration and retention (EPR) effects [39,40], thereby significantly increasing the local drug concentration at the tumor site. In other words, the presence of red blood cell membranes may enhance endocytosis and allow more drugs to enter cancer cells to achieve effective antitumor effects. 

During the administration process, the results of the relative body weight change of the mice are shown in Figure 5d. The relative weight of the mice treated with RBC-NC(DOX) did not show a specific downward trend, and there was no obvious abnormality in the weight change, reflecting the overall safety of the formula. However, for the free DOX and NC(DOX) treatment groups, the observed weight change reflects the increase in tumor burden during the period, indicating a toxic effect on mice. During the treatment period, the weight of the RBC-NC(DOX) treated mice continued to increase, and the tumor burden was small, indicating that such bio-smart materials have the advantage of reducing the toxicity of the drug system. In future exploration, the nanoparticle formulation can be further studied to exert a stronger antitumor effect.

We prepared tissue sections of the tumor, heart, liver, spleen, lung, and kidney of mice in each treatment group, and the results of H&E staining are shown in Figure 5e–h. There are many methods for detecting apoptosis, and H&E staining is a commonly used morphological detection method. This method is simple and easy to implement for the preliminary observation of cell apoptosis and can be used as one of the analysis indicators [41,42]. H&E is an acid–base affinity dye. After staining, apoptotic cells will be darker, while the light red cytoplasm will overflow. In this way, apoptotic cells can be partially observed by H&E staining. As shown in Figure 5h, the cell color of the tumor tissue in the RBC-NC(DOX) group was darker than that in the other groups, indicating that compared to free DOX and NC(DOX), RBC-NC(DOX) has a better killing effect on tumor cells. The intercellular spaces of tumor tissues in the RBC-NC(DOX) group were more obvious than the spaces between the normal saline group and other treatment groups, and the shrinkage was obvious. It was observed that compared with the normal saline group, the tissue sections of the heart, spleen, lung, and kidney of the mice in the other treatment groups were not very different. It shows that the carrier and the drug have no obvious toxicity to these five organs of mice. After comparison, it was found that the liver tissue slices of the free DOX group and NC(DOX) had obvious damage, and the RBC-NC(DOX) treatment group did not observe obvious liver tissue damage, indicating that doxorubicin caused damage to the liver tissue of the mice. It is further proved that the application of the cell membrane-encapsulated nano-drug delivery system reduces the kidney toxicity of the drug.

### 3.6. Blood Compatibility

Our synthetic drugs enter the human body through intravenous injection and first contact with the blood. Therefore, we must evaluate the biocompatibility of the drug through the interaction between the drug and the blood. Figure 6a is a comparison chart of the hemolysis rate of free DOX, NC(DOX), and RBC-NC(DOX). From left to right in Figure 6b are the positive control, negative control, free DOX, NC(DOX), and RBC-NC(DOX) respectively. The drug concentration is 50 μg/mL. This rule is intuitively expressed with photos. The hemolysis rate of the RBC-NC(DOX) group was the lowest, which proves that RBC-NC(DOX) has better biocompatibility than the preparations without red blood cell membrane.

## 4. Conclusions

The above research shows that the RBC-NC platform we prepared has advantages that other traditional drug carriers cannot match: extending the circulation time, enhancing stability, and improving the efficacy of drugs on tumors. Importantly, under the conditions of intravenous injection, the natural-derived cell membrane coating system provides better biocompatibility and reduces side effects. In addition, the existence of pure drug nanocrystalline cores enables a large number of hydrophobic chemotherapeutic drugs to be safely and reliably delivered to the tumor site. It can be seen that functionally inner cores (such as active small molecules, enzymes, nucleic acids, etc.) provide the possibility to achieve more therapeutic functions [43]. Although the source and storage of erythrocyte membrane carriers need to be further studied, considering the existing blood transfusion infrastructure and in vitro blood cell culture technology [44,45], it can be synthesized and used on a large scale in the future.

In general, this potential delivery strategy can deliver a variety of low solubility or high toxicity drugs (such as paclitaxel, docetaxel, and irinotecan) through the RBC-NC platform to treat a wide range of cancers [46,47]. However, the mechanism and conditions of the combination of the red blood cell membrane and the different inner cores need to be further explored. Coating larger size nanoparticles in the red blood cell membrane and reducing the damage to the red blood cell membrane are also problems that need to be solved. Finally, the erythrocyte membrane drug delivery platform can be further optimized to improve the targeting and controlled release of the treatment. The content verified in this article provides a promising foundation for the continued development of RBC transmission systems, future clinical trials, and nanomedicine.

## Figures and Tables

**Figure 1 nanomaterials-11-02513-f001:**
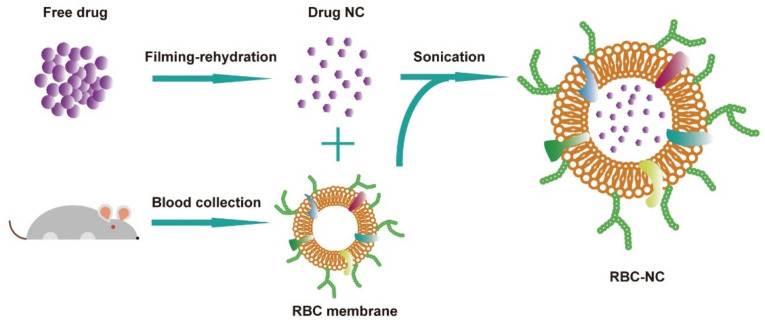
Schematic depicting the preparation of doxorubicin (DOX)-loaded RBC, denoted “RBC-NC(DOX)”.

**Figure 2 nanomaterials-11-02513-f002:**
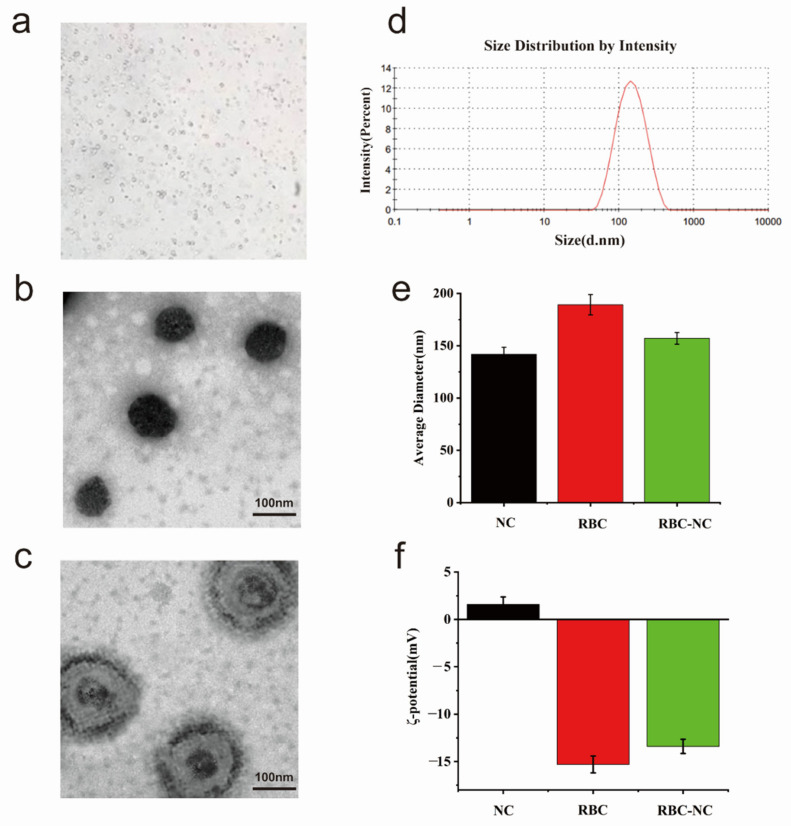
(**a**) The morphology of RBC-NC(DOX) observed through the microscope. (**b**) Transmission electron microscopy (TEM) visualization of NC(DOX) and (**c**) RBC-NC(DOX) with phosphotungstic acid staining (scale bar = 100 nm). (**d**) Particle size distribution of RBC-NC(DOX). (**e**) The changes of particle size before and after RBC package. (**f**) The changes of potential before and after RBC wrapping. Error bars represent standard deviations (n = 3).

**Figure 3 nanomaterials-11-02513-f003:**
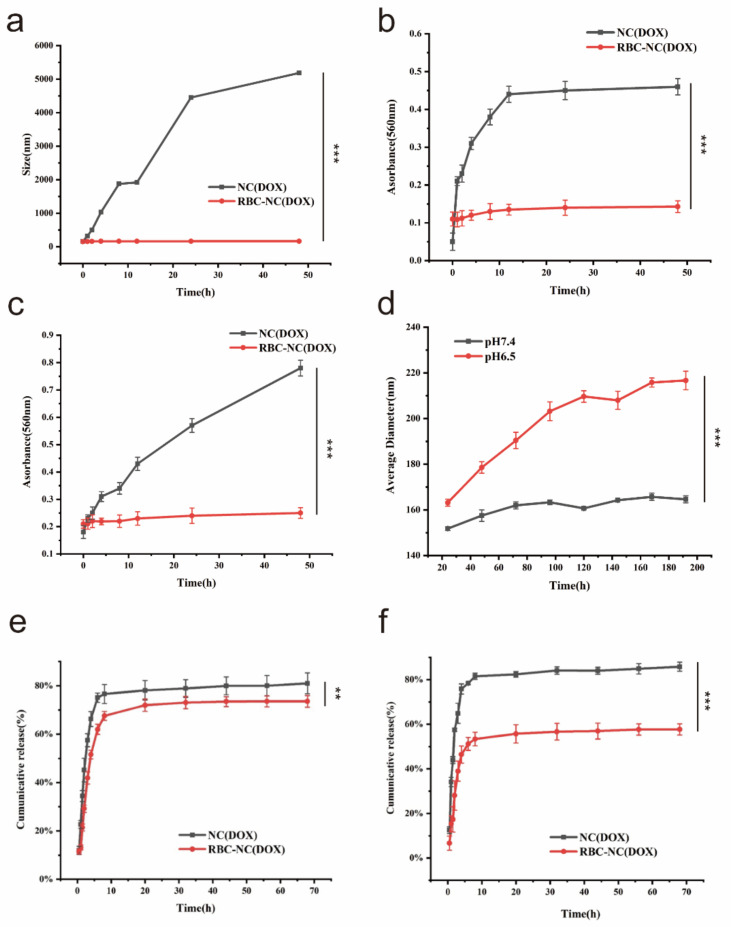
(**a**) Long-term stability study of NC(DOX) and RBC-NC(DOX) in 2 days. (**b**) The absorbance of NC(DOX) and RBC-NC(DOX) at 560 nm in PBS and (**c**) 10% FBS. (**d**) The size change of RBC-NC(DOX) in different environments. (**e**,**f**) Cumulative release profile of DOX from NC(DOX) and RBC-NC(DOX). (**e**) pH 6.5, (**f**) pH 7.4. Error bars represent standard deviations (n = 3). ** and *** indicate *p* < 0.01 and *p* < 0.001, respectively.

**Figure 4 nanomaterials-11-02513-f004:**
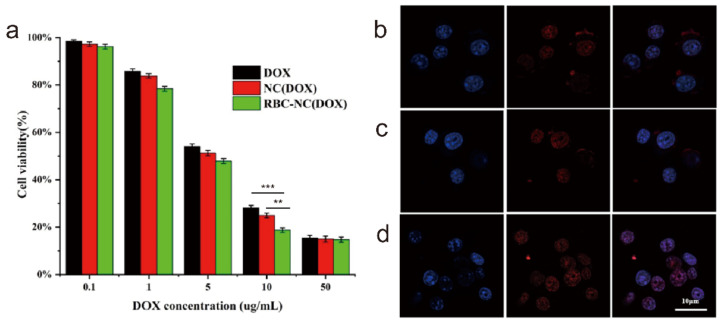
(**a**) The cytotoxic effects of different DOX formulations on 4T1 cells. (**b**–**d**) The uptake of different DOX preparations by 4T1 cells using a confocal laser scanning microscope. (**b**) Free DOX, (**c**) NC(DOX), (**d**) RBC-NC(DOX). Error bars represent standard deviations (n = 3). ** and *** indicate *p* < 0.01 and *p* < 0.001, respectively.

**Figure 5 nanomaterials-11-02513-f005:**
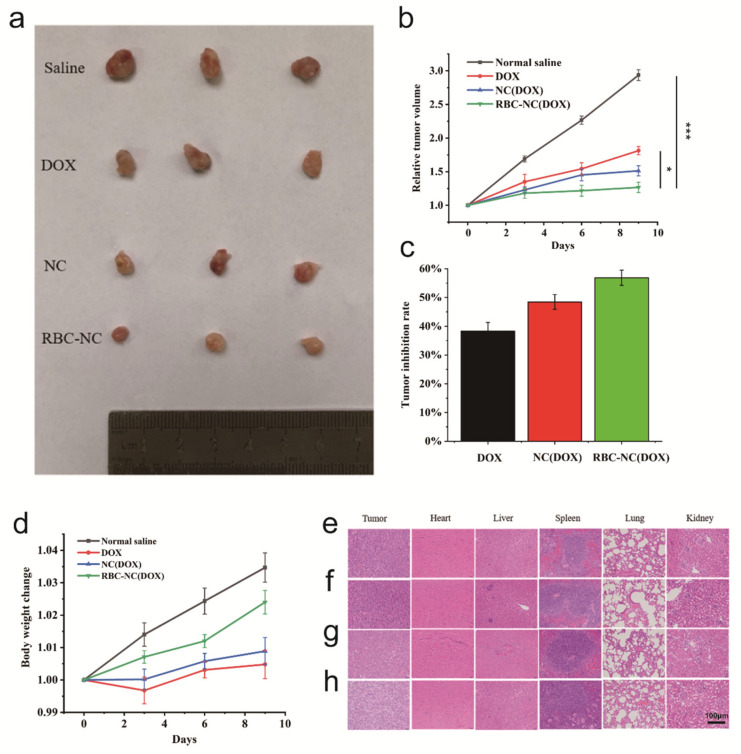
(**a**,**b**) Actual tumor morphology changes. (**c**) Tumor growth inhibition in mice treated with normal saline, free DOX, NC(DOX), and RBC-NC(DOX) by tail vein injection. (**d**) Body weight change over time. (**e**–**h**) H&E-stained histological sections. (**e**) Normal saline, (**f**) Free DOX, (**g**) NC(DOX), and (**h**) RBC-NC(DOX). Error bars represent standard deviations (n = 6). * and *** indicate *p* < 0.05 and *p* < 0.001, respectively.

**Figure 6 nanomaterials-11-02513-f006:**
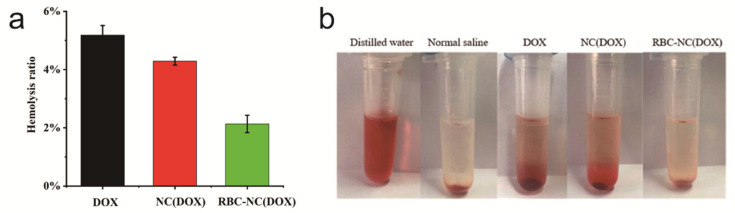
(**a**) The chart and (**b**) photo of the hemolysis of free DOX, NC(DOX), and RBC-NC(DOX). Error bars represent standard deviations (n = 3).

**Table 1 nanomaterials-11-02513-t001:** IC_50_ values of different formulations against 4T1 cells.

Formulation	IC_50_ (μg/mL)
DOX	5.587
NC(DOX)	4.812
RBC-NC(DOX)	3.805

## Data Availability

The data presented in this study are available.

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
