# Peer review of "Preparation and Characterization of Anti-Cancer Crystal Drugs Based on Erythrocyte Membrane Nanoplatform"

_nanomaterials, 2021, doi:10.3390/nano11102513_

Round 1
Reviewer 1 Report
This is a very interesting study from Ren et al. characterizing RB-based nanocrystals and evaluating their antitumor effect in vitro and in vivo.
However, I have a couple of comments justifying major revision of your manuscript.
1) It is not mentioned how many times the assessment of potential and particle size, the stability, the aggregation, hemolysis and the in vitro DOX release were performed.
2) It is quite surprising that no statistical analysis was performed to compare the different formulations. The absence of statistical analysis makes the manuscript conclusions quite poorly relevant and not acceptable for such a scientific journal like Nanomaterials.
3) Fig. 3 : please provide a revised Fig. 3a with not truncated y axis. Please also change Fig. 3d legend color as it is not the same as the other parts of the graph.
4) Antitumor effect is evaluated on the sole basis of tumor size. A complementary study should be performed on histological sections, for instance investigating at least apoptosis and angiogenesis.
5) Please send your manuscript to an English proofreading service as there are many typos in the actual manuscript.
6) in vivo experiments:
6.1) mat&meths dramatically lack an ethical statement regarding animal experimenting (ethics committee, project authorization number...).
6.2) Please be more accurate and precise about the route of administration, the injection volume, the injection conditions (anesthesia or not ?) of DOX and derivatives.
6.3) Please justify the choice of immunocompetent BALB/c mice for a tumor model. Please precise the origin of 4T1 cells, justify the choice of this lineage, and discuss the limits of evaluating your formulations on this sole tumor model.
6.4) Globally, the in vivo results are poorly discussed with no reference to the literature. This should be critically enhanced.
Author Response
请参阅附件。

Reviewer 2 Report
Authors report on use of red blood cell membranes (or RBC ghosts?) for delivery of doxorubicine prepared as nanocrystalline solids for application in cancer therapy. This is an interesting idea confirmed by very promising results. Such material is of interest to many readers and deserves to be published.
The idea is well documented by the results. However, interpretation of some is misleading and some experimental details are not clear, e.g.
Line 57: “…treatment of nanocrystals against tumors is based on prolonged circulation of drug …” semantics and logic of language, use rather “…treatment of tumors by nanocrystalline drugs”
Line 84: “When the dichloromethane in the solution is evaporated to dryness through rotary evaporation at 40°C, and then we vacuum dry the rest for 2h. Next, the resulting solution is added to 1 mL of distilled water…”
How come that we still have DOX solution after evaporation to dryness? If yes, how amount of DOX has been evaluated? Howe much of DOX contains 1 mL of distilled water? How samples used for testing were prepared?
Line 110:”We take 5ml solutions of different DOX preparations and dialyze them in 50mL of…” What means “different DOX preparation”? Be more precise.
Line 173: “The general structure of the generated nanoparticles is shown in Figure 1a, the DOX crystals are loaded in the core, and the RBC membrane coating and all related proteins form the outer layer.”
According to Figure 2c this not true. DOX is contained not only inside of the RBC ghosts but also included in the membrane. There is only one remark on this point in line 254: “…the red blood cell membrane is coated,…”. In the whole text these nanoparticles are treated exclusively as DOX containers.
Therefore, further discussion should be remodeled accordingly.
Line 200: In order to continue to characterize the RBC-NC (DOX) we synthesized that whether the red blood cell membrane is covered with doxorubicin crystals, we measured the nanoparticle size of NC(DOX) and RBC with a nanoparticle size analyzer.
It does not make much sense.
Line 248: “…all prescriptions. Figure 3e-f are the …”, probably preparations(?)
Line 274: “…we can visually judge the uptake of different DOX preparations by 4T1 cells …”
Probably “evaluate” would be better
Finally, authors used mice for their in vivo research. This raises problem of ethical issues that should be obligatory clarified.
Round 2
Reviewer 1 Report
The manuscript is now improved.
Still, there are missing statistics in Fig. 5 and Fig. 6 (results being described as significant in the manuscript but not shown on the figures).
p. 12 l. 369: Please use the word "significantly" for commenting on statistical results only. Comparing histological sections visually can not lead to a significant result unless you provide a scoring grid or something equivalent that underwent statistical analysis.
It is quite a shame the authors did not consider adding an experiment to answer point 4 (TUNEL or caspase). The discussion and conclusions on antitumor efficacy should be balanced with the perspective of a future experiment dosing DOX in the tumor and/or following the biodistribution of the nanocrystals and evaluating the induced apoptosis. I'm still not convinced by the argumentation added in the discussion.
The ethical authorization number was written in the authors' response but not in the manuscript.
Reviewer 2 Report
Article is acceptable in its present form.
